# Characterizing Complex Mineral Structures in Thin Sections of Geological Samples with a Scanning Hall Effect Microscope

**DOI:** 10.3390/s19071636

**Published:** 2019-04-05

**Authors:** Jefferson F. D. F. Araujo, Andre L. A. Reis, Vanderlei C. Oliveira, Amanda F. Santos, Cleanio Luz-Lima, Elder Yokoyama, Leonardo A. F. Mendoza, João M. B. Pereira, Antonio C. Bruno

**Affiliations:** 1Department of Physics, Pontifical Catholic University of Rio de Janeiro, Rio de Janeiro 22451-900, Brazil; acbruno@puc-rio.br; 2Department of Geophysics, Observatório Nacional, Rio de Janeiro 20921-400, Brazil; andrelreis@on.br (A.L.A.R.); vanderlei@on.br (V.C.O.J.); 3Department of Physics, University of California, Santa Barbara, CA 93106, USA; afariasdossantos@ucsb.edu; 4Department of Physics, Campus Ministro Petrônio Portella, Universidade Federal do Piauí, Teresina 64049-550, PI, Brazil; cleanio@ufpi.edu.br; 5Institute of Geosciences, University of Brasília, Brasília 70910-900, Brazil; eyokoyama@unb.br; 6Department of Electrical Engineering, Universidade Estadual do Rio de Janeiro, Rio de Janeiro 20550-900, Brazil; mendonza@ele.puc-rio.br; 7Fiber Optics, RISE Acreo, Electrum 236, 164 40 Kista, Sweden; joao.pereira@ri.se

**Keywords:** magnetic scanning microscope, hall sensor, magnetic materials, geological sample

## Abstract

We improved a magnetic scanning microscope for measuring the magnetic properties of minerals in thin sections of geological samples at submillimeter scales. The microscope is comprised of a 200 µm diameter Hall sensor that is located at a distance of 142 µm from the sample; an electromagnet capable of applying up to 500 mT DC magnetic fields to the sample over a 40 mm diameter region; a second Hall sensor arranged in a gradiometric configuration to cancel the background signal applied by the electromagnet and reduce the overall noise in the system; a custom-designed electronics system to bias the sensors and allow adjustments to the background signal cancelation; and a scanning XY stage with micrometer resolution. Our system achieves a spatial resolution of 200 µm with a noise at 6.0 Hz of 300 nT_rms_/(Hz)^1/2^ in an unshielded environment. The magnetic moment sensitivity is 1.3 × 10^−11^ Am^2^. We successfully measured the representative magnetization of a geological sample using an alternative model that takes the sample geometry into account and identified different micrometric characteristics in the sample slice.

## 1. Introduction

Scanning magnetic microscopy has attracted a significant amount of interest in recent years. It has the potential to elucidate a number of problems in science and engineering that involve magnetization images [1,2,3,4]. The applications involving scanning magnetic microscopy cover many disciplines, ranging from physics and materials science [5,6,7,8] to paleomagnetism, magnetism [9,10] and biophysics [11,12,13]. For example, in paleomagnetism, scanning magnetic microscopy can be used to recover information about past magnetic fields by analyzing the magnetizations of terrestrial or extraterrestrial rocks. Such rocks or meteorites can hold magnetic information for millions or even billions of years. However, the record of the oldest magnetic fields is commonly superimposed by other magnetic records acquired during the geological history of the material. Classic paleomagnetic techniques are based on the external measurement of the sample’s magnetic field. Thus, they reflect the vector sum of the magnetic moments of all rock minerals by measuring the bulk magnetization of a cylindrical sample. In particular, the rocks that have developed complex structures, such as breccia rocks, folded rocks and sedimentary rocks, may be particularly difficult to analyze using conventional paleomagnetic techniques [14]. In addition, these complex structures often have multiple magnetic mineral phases. Although these phases can be identified by usual rock magnetism techniques (e.g., curves of hysteresis cycles from bulk samples), it is usually difficult to spatialize and correlate these phases in relation to their magnetization [15].

Scanning magnetic microscopes are a new class of magnetometers that are capable of producing magnetic field maps with a spatial resolution of a few submillimeters. Broadly, a sample is moved horizontally under a tiny magnetic sensor that is located very close to the surface of the sample. After this, a magnetic map is constructed by recording the measurement of the magnetic field at each point in a regular grid of positions [1,2,3,4,5,6,7,16,17,18]. Additionally, the majority of magnetic microscopes operate at low temperatures because they use SQUID (superconducting quantum interference device) sensors and magnetic shielding due to the sensitivity of the reading systems [16,17,18].

With this objective, we developed a magnetic microscope that can perform magnetic mapping at ambient temperatures with a large scanning area; has the ability to handle samples on the millimeter scale; and can identify distinct constituents (i.e., different magnetic mineral phases) of complex structures. The microscope has a small electromagnet that can apply a uniform DC magnetic field of up to 500 mT in a direction that is perpendicular to the sample. In the current configuration, the scanning magnetic microscope is equipped with a pair of low-cost commercial Hall effect sensors, thus forming an axial gradiometer. The gradiometer serves a dual purpose: canceling the magnetic field applied by the electromagnet and reducing the ambient magnetic noise since it operates in an unprotected environment. The achieved sensitivity of the magnetic moment (1.3 × 10^−11^ Am^2^) can allow the study of geological samples, solid samples, liquids, microstructures and nanostructured samples. Furthermore, the spatial resolution of the current sensor (200 μm) can allow the measurement of magnetic maps in different regions of the same sample. Through these maps, the sample is characterized in different positions, showing that there are different minerals within the same sample in the case of geological samples.

## 2. Scanning Magnetic Microscope

### 2.1. Mechanical Design

The microscope allows the scanning of thin sections of rock samples up to 40 mm in size, which are placed between the poles of an electromagnet (3470, GMW Inc.). The electromagnet is capable of generating DC magnetic fields of up to ~500 mT over a 40 mm diameter area with a 3.5 A pole gap current. The electromagnet is positioned such that its pole axes are oriented in the vertical direction (Figure 1a). The sample is placed upside down in the sample holder using a double-sided adhesive tape. To sense the response generated by the sample to the applied field, we use two commercial Hall sensors, HQ-811 (AKM, Corp), henceforth designated Sensor A and Sensor B, which incorporate an InAs element in a SMT package (Figure 1b). Their 200 μm diameter sensing areas have a nominal distance of 130 μm to the top surface. The two sensors are connected in an axial gradiometric configuration and fixed on opposite sides of a 2.2 mm thick printed circuit board (PCB).

As shown in Figure 2a, the sample holder is connected to a micropositioner (XY Stage) that is attached to a support base. Sensor A near the sample was glued to the PCB using transparent epoxy resin. To get closer to the sample, Sensor A is lapped until its 4 connection terminals appear on the top surface (inset in Figure 2b). Sensor B acts to reduce the field generated by the electromagnet in the output signal. Sensor C measures the applied field (Figure 2b). We estimate that the new distance between the sensor and the top surface is 100 μm. Further lapping will prevent the sensor from working properly. The PCB is mounted on an acrylic structure that is fixed to one of the electromagnetic poles. For Sensor C, a MLX-90215 programmable linear Hall sensor was used and glued to the acrylic support.

### 2.2. Custom Electronics

We can bias the HQ-811 sensors using a certain current or voltage. After several tests, we concluded that biasing using a current in the 1–5 mA range produced the best signal-to-noise ratio. The circuit built consisted of two independent current sources and instrumentation amplifiers for signal amplification and decoupling (see Figure 3a). A low-noise preamplifier (SR560, SRS Inc. Sunnyvale, CA, USA) carried out the gradiometer operation by electronically subtracting the two output signals. In the first version, the current sources were based on the IC LM334 and were controlled by resistance. However, this produced high noise and had a strong temperature dependence even when using a temperature compensation circuit [19]. Following this, for the two current sources that are controlled by voltage, we redesigned the circuit by replacing the LM334 with the INA105. This redesign provided better results. Figure 3b shows a comparison between the noise of the two custom electronics.

We used a biasing AC current at a 1.0 kHz frequency and a peak amplitude of 1.0 V. For comparison purposes, we added a magnetic signal at 4 Hz with a peak amplitude of 5 µT to the measurements. The spectrum indicated by the legend INA105 shows a noise level of approximately 300 nT_rms_/(Hz)^1/2^ at 6 Hz. We primarily designed the gradiometer for rejecting the applied field, increasing the dynamic range of the instrument and allowing its operation at the high applied fields. However, the gradiometer also rejects the ambient magnetic noise. The applied field rejection was tuned by applying a uniform field of 500 mT before adjusting the biasing current of Sensor B until the output read 0.5 mT, which is a factor of 1.000 in field rejection. Afterwards, to evaluate the effect on the magnetic noise, we measured a sample consisting of a cavity that had a diameter of 500 µm and a depth of 400 µm and was filled with 99.9% pure magnetite (Fe_3_O_4_) fine particles. A mass of 102 µg was used. Due to the proximity of the sample port to the sensors, the best model to fit the experimental curves is currently a cylinder because the cavity where the sample is deposited has the shape of a cylinder. Assuming that our sample consists of fine particles and completely fills the cavity, it has a magnetic moment (m_z_), which is uniformly distributed in the z-direction (the magnetic field applied by the electromagnet H has the same direction); a radius *a*; and a complement of 1. The z-component of the field can be obtained using the Biot–Savart law [20,21]. By measuring B_z_ for several values of the applied field and using (x*′*, *y′*, *z′)* that were obtained with the aid of an optical microscope and the sensor plug, it was possible to estimate these relative distances between the center of the sample holder and the center of the active part inside the sensor. Therefore, we can obtain the magnetic moment as a function of the applied field according to the equation below [21]:(1)$$m_{z}\left(H\right)=\frac{4\pi^{2}a^{2}l}{\mu_{0}\int_{-l/2}^{l/2}\int_{0}^{2\pi }\frac{\left(z^{\prime }-z\right)a\;\cos\;\phi }{r^{\prime^{3}}}d\phi dz}B_{z}\left(x^{\prime },y^{\prime },z^{\prime }\right)$$
$r^{\prime }$, which is the distance between the elements, is defined as $r^{\prime }={\left[{\left(x^{\prime }-x\right)}^{2}+{\left(y^{\prime }-asen\varphi \right)}^{2}+{\left(z^{\prime }-a\cos\varphi \right)}^{2}\right]}^{1/2}$, where $d\vec{l}$ is the distance between the element and point p in space XY and $d\vec{l}=ad\varphi \;\cos\varphi \;\vec{j}$ while µ_0_ is the permeability of free space and the radius of a cylinder that is uniformly magnetized along its length *l*. We can determine the real *z* distance between the sample and the sensor by scanning a line at the top of the sample *B_z_* = (*x′*, *y′*, *z′*) and analyzing only the spatial dependence of *B_z_* with *a* = 250 µm and *l* = 400 µm. Using a least-squares fitting routine, we obtained 142 μm for the actual distance from the Hall sensor element to the top of the sample. Using this distance and 5197 kg/m^3^ for Fe_3_O_4_ bulk density, we found that the magnetization was 75.8 Am^2^/kg at 500 mT, which was approximately 0.19% higher than the value obtained with a vibrating-sample magnetometer (VSM) and 0.23% lower than the value obtained with a Hall magnetometer [22,23,24].

We compared the gradiometer and top sensor (Sensor A) outputs. As shown in Figure 4, the results are from the remanent field acquired after a magnetization of 500 mT. Figure 4a shows a 5 × 3 mm map of the remanent field using the gradiometer. In Figure 4b, the same scan with only Sensor A shows a degradation in the signal-to-noise ratio. In Figure 4c, the centerline (*y* = 1.5 mm) of the two maps are shown together with a theoretical model of the magnetite particles that have an estimated magnetic moment of 6.8 × 10^−7^ Am^2^. The gradiometer action of reducing the ambient noise is even more evident.

## 3. Measurement of Geological Samples

### 3.1. Microscopy of Geological Samples

To demonstrate the characterization abilities of our magnetic microscope, we mapped a geological sample that is typical of the ones found in the Jack Hills. The Jack Hills are a 70 km long range of hills located on the southern margin of Narryer Terrane, Western Australia [25]. It comprises an Archean–Paleoproterozoic greenstone belt that is surrounded by granitic and gneissic rocks. The greenstone belt is a sequence of metavolcanic and metasedimentary rocks that usually have a polycyclic history of metamorphism and deformation. A small part of the Jack Hills greenstone belt is composed of banded iron formation (BIF) that records a significant proportion of the deformation history [26]. According to Spaggiari, this BIF can preserve at least three generations of folding deformation that can be observed at both the meso- and microscales.

To demonstrate the capability of our microscope, we scanned a polished section of Jack Hills BIF sample. This sample shows a microfold of alternating microbands of iron oxyhydroxides and silica (see Figure 5a). Figure 5b shows the perpendicular component of the magnetic field response to a 20 × 25 mm scan of the microfold sample in the presence of a 500 mT field that is applied perpendicularly to the sample and observed at three points. The first point in blue is located at *x* = 43.7 mm and *y* = 35 mm; the second point in green is located at *x* = 45.2 mm and *y* =38 mm; and the last third point in black is located at *x* = 46.8 mm and *y* = 40 mm.

Figure 5c shows three curves of a hysteresis cycle at the three different fixed points of the same sample shown in Figure 5b. The curves represent the magnetic response of the induced field of the sample, which was obtained by positioning the sample at the points shown in Figure 5b in relation to the sensitivity axis of the microscope reading system set in the position of the red point. We observed that these curves are different. Therefore, they represent different minerals within the same sample. This type of study cannot be performed with conventional magnetometers because the sample is treated as being evenly distributed [20,22,23,24]. To verify the presence of different minerals, we performed Raman spectroscopy analysis of a small region of the sample (see Figure 5a,d). Raman measurements were performed using a micro-Raman Senterra Bruker spectrometer and the 785 nm line of a laser was used as the excitation source. The spectrometer slit was set for a resolution of 4 cm^−1^. An optical microscope (Olympus BX-50) with an Olympus MPlan N 20×/0.40 NA objective was used to focus the laser on the sample surface and to obtain the images (Figure 5d). In this figure, we selected a rectangle to mark a new region, which results in small circles that are numbered from 1 to 4. These circles show where the Raman spectra of the sample identified as Point 1, Point 2, Point 3 and Point 4 were obtained. When we analyzed the obtained spectra, we observed that the predominant concentration in the sample is of hematite and quartz. This assertion is because the characteristic peaks of these phases are observed in the other spectra but in addition to these phases, the goethite and magnetite phases were also observed.

In addition to Jack Hills BIF, we scanned a sample of the metamorphic rock with a diameter of 300 km from the Vredefort Dome in South Africa. The Vredefort Dome is a 90-km-wide central uplift of a 300-km-wide eroded impact structure e.g., [27]. The exposed Vredefort central uplift is comprised of the polydeformed Archean migmatitic gneisses and granitoids with a scattered occurrence of metasedimentary and mafic granulite xenoliths [27]. The sample in the form of a 30 μm thin section (see Figure 6a,b) was prepared from a core drilled in granulite gneiss that has dimensions of 9 × 9 m^2^ and the dark regions contain magnetic carriers and are surrounded by nonmagnetic plagioclase feldspar and quartz [28]. Figure 6b shows a picture taken with an optical microscope of a small region of the thin section that is denoted the ‘hook’ in the picture of the slice.

### 3.2. Modeling Using Current Circuit

We subjected the sample to magnetic fields varying from 415 mT to −31 mT. Figure 7 shows a set of 15 experimental B_z_ maps (25 µm step) of the hook for each applied magnetic field. From the maps, a weak contribution from the upper part of the hook is observed, which we termed the ‘head’. The main body (‘handle’) of the hook presents a stronger magnetic field response. A model is needed to characterize these two regions of the sample. The model of a single dipole is useful when the sample is distant or when it has a spherical shape. This is not the case with this microscope because the sensor is very close to the sample. For this case, the current circuit model was developed with assumption of both the head and handle. The calculation starts from the Biot−Savart law (see Equation (2)):(2)$${\vec{B}}_{z}\left(x^{\prime },y^{\prime },z^{\prime }\right)=\frac{\mu_{0}}{4\pi }\int_{loop}\frac{Id\vec{l}}{r\prime^{2}}\times \frac{{\vec{r}}^{\prime }}{r^{\prime }}$$
where *I* is the current in the circuit and *dl* is the element of length on which the integral along the area of the sample is calculated. The induced field of the sample in the presence of the applied field is acquired directly from the map, which is measured in the scanning magnetic microscope. Therefore, by knowing the B_z_(*x′*, *y′*, *z′*) and taking the area of the sample, we can extend the value of the current using Equation (3). If the current is confined in an XY plane and the reading is in the z-direction, the magnetic moment can be represented as follows:(3)$$m_{z}=I\left(Area\right)$$

It is important to note that Equation (3) does not consider the shape of the circuit. By knowing the area of the current circuit, we can estimate the value of the magnetic moment m_z_ in Am^2^.

To quantify these two main contributions to the detected magnetic field, we used the two current loop models in the shape of the hook as shown in Figure 8a. We called the top loop the “head” and the bottom loop the “handle”. Figure 8b shows the simulated perpendicular component of the field for a 415 mT applied field. It is important to note that the B_z_ in the model coincides with B_z_ as measured experimentally (see Figure 7; ~415 mT).

To find the current in each loop that adjusts the magnetic field generated by the model to the experimental field, we used the field along the dotted line (Z = 1100 µm) for the “head” and the field along the dashed line (Z = 350 µm) for the “handle”. Figure 8a,c and d show the perpendicular magnetic field component at each of the lines for the different values of the applied field in the handle and head regions, respectively. Figure 9a,b show the fitting results for an applied field of 415 mT using the model of a current circuit. The current values found a relationship between the area of each loop and the magnetic moment of each part of the hook. For the “handle”, $\mu_{s}=1.92\times 10^{-6}{\;\mathrm{Am}}^{2}$ and for the “head”, $\mu_{s}=1.51\times 10^{-7}{\;\mathrm{Am}}^{2}$.

Finally, Figure 9c shows the magnetization curves for each part of the hook, suggesting that these parts may be composed of different minerals.

### 3.3. Modeling Using Polygonal Prisms

To investigate the contribution of the two parts of the hook (i.e., head and handle), a different approach from that used in the previous section (current circuit) can be applied. This alternative approach is based on a methodology that is widely used in geophysics for modeling potential fields proposed by Plouff [29]. It calculates the potential field of a 3D prism with a polygonal cross-section.

Let **d**^o^ be an N-dimensional vector whose *i*th element is the vertical component of the magnetic field produced by a magnetic source in the position (*x^i^*; *y^i^*; *z^i^*) (Figure 10a). Considering that the sample can be approximated by a set of L polygonal prisms positioned according to a right-handed Cartesian coordinate system and considering that the *x*-, *y*- and *z*-axes are positively oriented to the north, the east and downward, respectively, we assume that each prism represents a different homogeneously magnetized region, with the edges coinciding with the bounds of the hook. We can estimate the magnetic moment m*^k^*, *k* = 1, …, *L*, by comparing the synthetic data produced by the model and the vertical component of the magnetic field map measured by magnetic microscopy since the sensor-to-sample distance, the thickness of the thin section and the magnetization direction are known. Mathematically, the vertical component of magnetic field B*_z_* produced by a set of polygonal prisms at point (*x^i^; y^i^; z^i^*) is given by:(4)$${\vec{B}}_{z}\left(x^{i},y^{i},z^{i}\right)\equiv B_{z}^{i}=\overset{L}{\underset{k=1}{\sum }}b_{z}^{\mathrm{ik}}\left(x^{i}{,y}^{i},\;z^{i},\;{\vec{x}}^{k},\;{\vec{y}}^{k},\;{\hat{m}}^{k},\;m^{k},\Delta z\right)$$
where $b_{z}^{\mathrm{ik}}$ represents the effect of the *k*th prism at the *i*th point (*x^i^; y^i^; z^i^*); x*^k^* is a vector containing the *x*-coordinates of the vertices of the *k*th prism; y*^k^* is a vector containing the *y*-coordinates of the vertices of the *k*th prism; ${\hat{m}}^{k}$ is a unit vector in the direction of magnetization; $m^{k}$ is the magnetization intensity and Δ*z* is the thickness of each prism. Mathematically, the vertical component of the magnetic field produced by the *k*th prism is given by the following expression:(5)$$b_{z}^{\mathrm{ik}}=C_{m}{\vec{M}}^{ik}{\hat{m}}^{k}m^{k}$$
where *C_m_* is a constant that is proportional to free space permeability and
(6)$$\vec{M}\;\left(x^{i}{,y}^{i}{,\;z}^{i}{,\;x\;\to }^{k}{,\;y\;\to }^{k},\Delta z\right)={\vec{M}}^{ik}={\left[\partial_{xz}\varphi^{ik}\partial_{yz}\varphi^{ik}\partial_{zz}\varphi^{ik}\right]}^{T}$$
where the scalar function $\varphi^{ik}$ is given by
(7)$$\varphi^{ik}=\underset{v^{k}}{∭}\frac{1}{r^{ik}}dv^{k}$$
and
(8)$$r_{}^{\mathrm{ik}}=\sqrt{{\left(x^{i}-\alpha^{k}\right)}^{2}+{\left(y^{i}-\beta^{k}\right)}^{2}+({\;z}^{i}-\gamma^{k}{)}^{2}}$$
where $\alpha^{k}$, $\beta^{k}$ and $\gamma^{k}$ are the Cartesian coordinates of an element inside the volume *v^k^* of the *k*th 3D prism with a polygonal cross-section. This modeling is solved using a Python library Fatiando a Terra [30]. As shown in Figure 7, the sample was magnetized vertically with a magnetic field intensity varying from 415 mT to −31 mT. To demonstrate the applicability of the method, we use the 415 mT map (Figure 10a). To estimate the two main contributions of the hook, we approximate the sample using three polygonal prisms with a thickness Δ*z* = 30 µm and the vertices shown in Figure 10b. We generate the vertical component of the magnetic field of the model (Figure 10c). As we note in Figure 10, the observed data and the synthetic data produced by modeling are very similar. The magnetic moment estimated for the head part (green prism in Figure 10b) is *m_head_* = 1.89 × 10^−7^ Am^2^ and the magnetic moment for the handle part (blue prism in Figure 10b) is *m_handle_* = 2.52 × 10^−6^ Am^2^.

These results have the same order of magnitude as the results obtained using a current circuit model (see Figure 9c). Therefore, we can notice that the results obtained using polygonal prisms are consistent with the results from current circuit modeling.

## 4. Conclusions

We demonstrated the ability of the scanning magnetic microscopy to operate at room temperature using low-cost Hall effect sensors to detect and map the vertical magnetization components of polished rock samples with good spatial resolution. The microscope has a spatial resolution of 200 µm with noise at 6.0 Hz of ≈300 nT_rms_/(Hz)^1/2^ in an unprotected environment and the sensitivity of the magnetic moment is 1.3 × 10^−11^ Am^2^ in the presence of a magnetic field of up to 500 mT. The microscope has a scanning range from 100 μm to 100 mm. The sensors were configured as axial gradiometers, which successfully reduced both the applied field and magnetic noise in the output signal.

We successfully measured the representative magnetization of a geological sample using an alternative model that takes the sample geometry into account and identified different micrometric characteristics in the sample slice. The model used was compared with another model using polygonal prisms, which resulted in very similar magnetization values. Furthermore, we were able to spatially separate the different magnetic minerals through their individual hysteresis curves, which may open up new possibilities for the investigation of magnetic mineralogy.

## Figures and Tables

**Figure 1 sensors-19-01636-f001:**
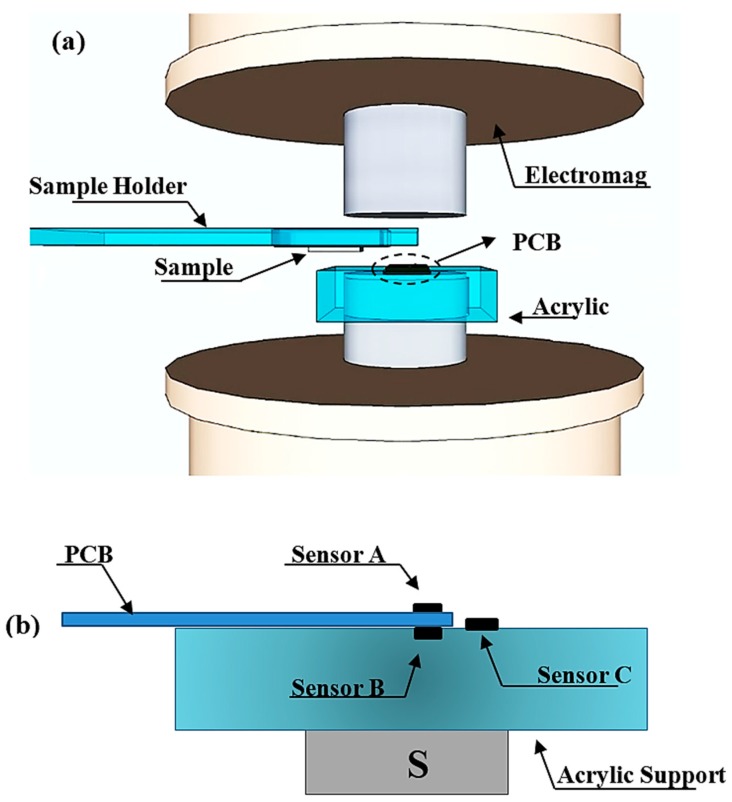
(**a**) Schematic drawing of the main components of the microscope: north and south pole of the electromagnet, an acrylic holder containing the PCB with Hall effect sensors, a sample holder that moves in the X and Y directions and the sample. Drawing is not to scale. (**b**) Detailed view of the PCB containing the gradiometric sensors (A and B) as well as an additional sensor (C) for measuring the applied field and the sample holder.

**Figure 2 sensors-19-01636-f002:**
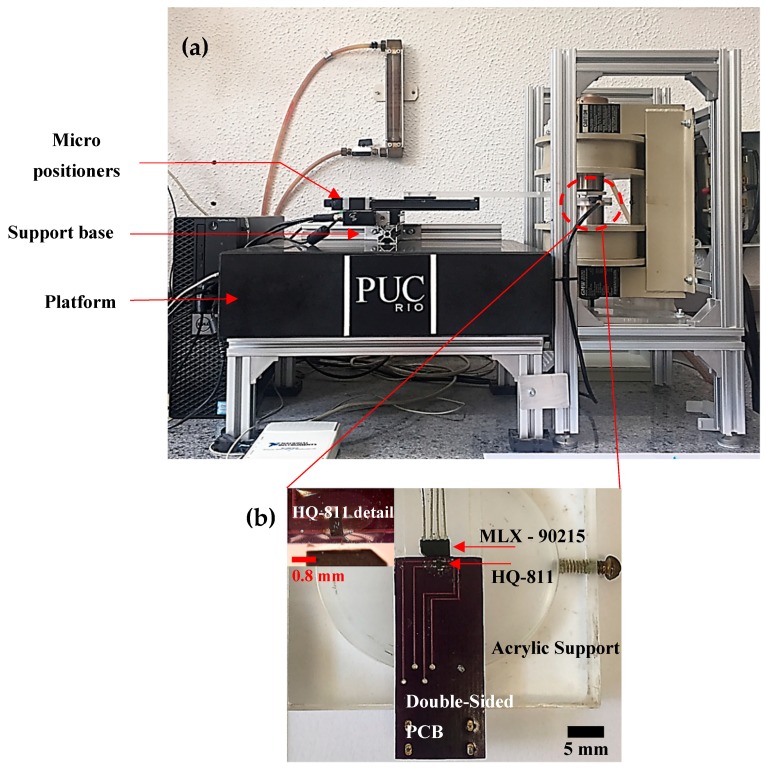
(**a**) Photograph of the magnetic scanning microscope. (**b**) Photograph of the Hall sensors and acrylic structure that fits around one electromagnetic pole. Two HQ-811 sensors on each side of the PCB form an axial gradiometer. The MLX-90215 sensor measures the applied field. At the top, we have the increment. We can check details of the connection of the HQ-811 photo sensor with a scale.

**Figure 3 sensors-19-01636-f003:**
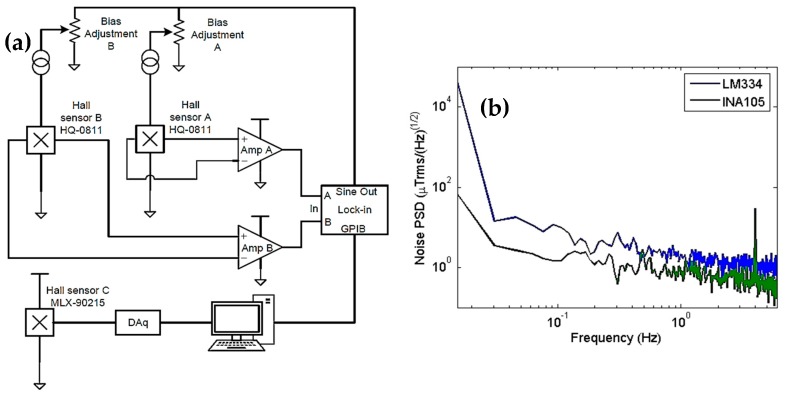
(**a**) Electronics using the INA105 schematics for current sources and preamplification for the Hall gradiometer. (**b**) Noise spectrum of the HQ-811 sensor with two custom electronics measured outside a magnetic shield in the laboratory. For comparison purposes, a magnetic signal at 4 Hz with a peak amplitude of 5 µT was added to the measurements. The noise level at 6 Hz is approximately 300 nT_rms_/(Hz)^1/2^.

**Figure 4 sensors-19-01636-f004:**
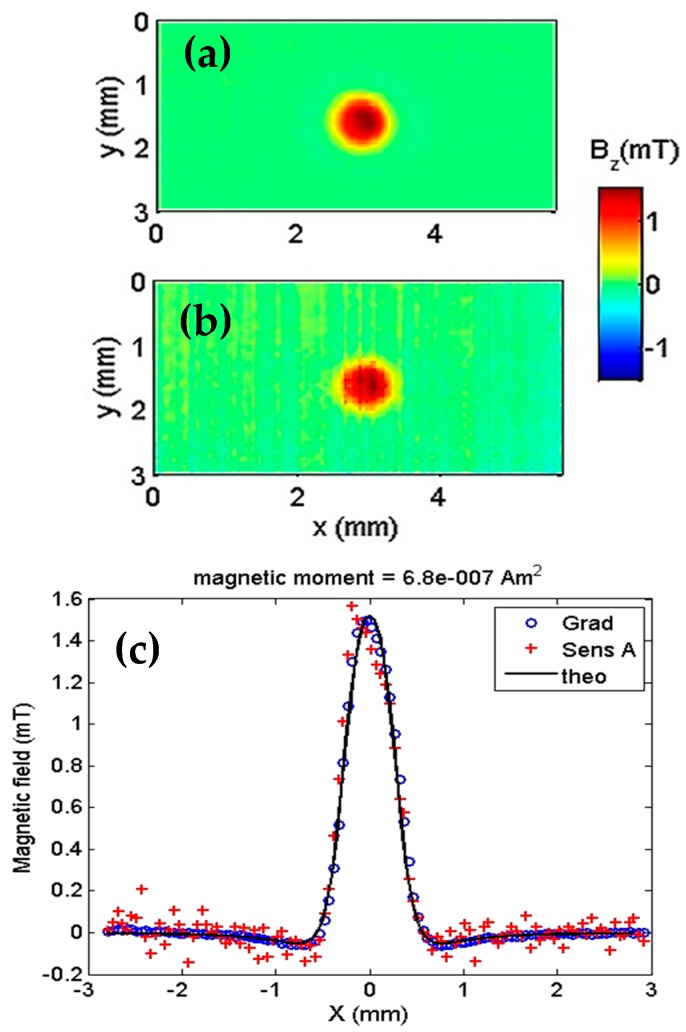
Gradiometer test with a sample consisting of a cavity with a diameter of 500 µm and depth of 400 µm and is filled with approximately 102 µg of 99.9% pure magnetite. (**a**) Map of the remanent field with the gradiometer turned on. (**b**) Map with only Sensor A. (**c**) Plot showing the theoretical field due to the magnetite sample (solid line), as measured by the gradiometer (circles) and measured by sensor A (crosses).

**Figure 5 sensors-19-01636-f005:**
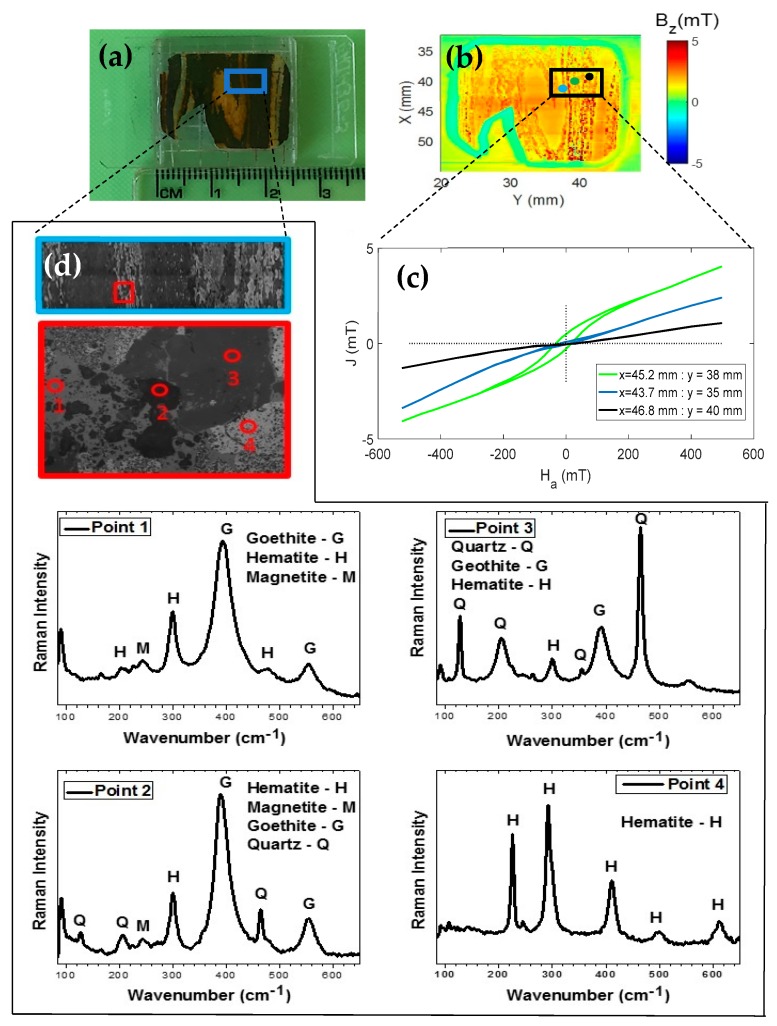
(**a**) Photograph of the microfold structure attached to the sample port. (**b**) The perpendicular component of the magnetic field response to a 20 × 25 mm scan of the microfold sample in the presence of a 500 mT applied field. We can see the three points. The first point in blue is located at *x* = 43.7 mm and *y* = 35 mm; the second point in green is located at *x* = 45.2 and *y* = 38 mm; and the third point in black is located at *x* = 46.8 mm and *y* = 40 mm. (**c**) Hysteresis cycle curves of the induced magnetic field J (mT) at the three points of the sample, which indicate that there are different minerals. (**d**) Images obtained from the optical microscopy of a line. Within this line, 4 points were identified at different positions and analyzed by Raman spectroscopy. The results of these points (1, 2, 3 and 4) led to the identification of different minerals in the same sample.

**Figure 6 sensors-19-01636-f006:**
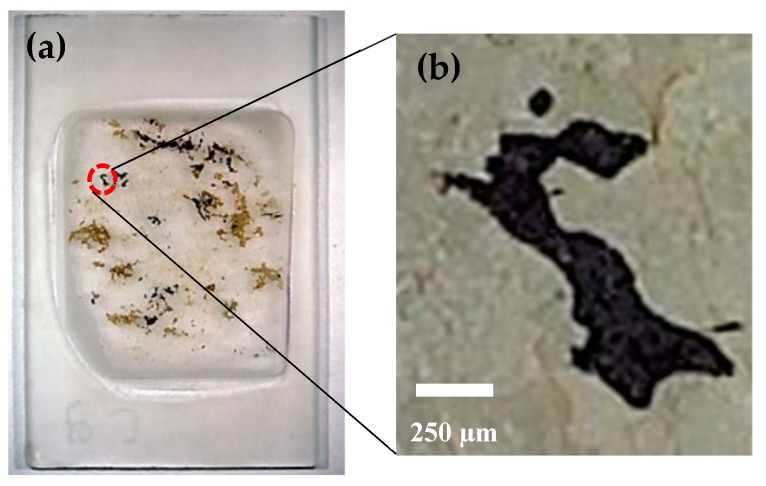
(**a**) Vredefort thin section. (**b**) A picture taken with an optical microscope.

**Figure 7 sensors-19-01636-f007:**
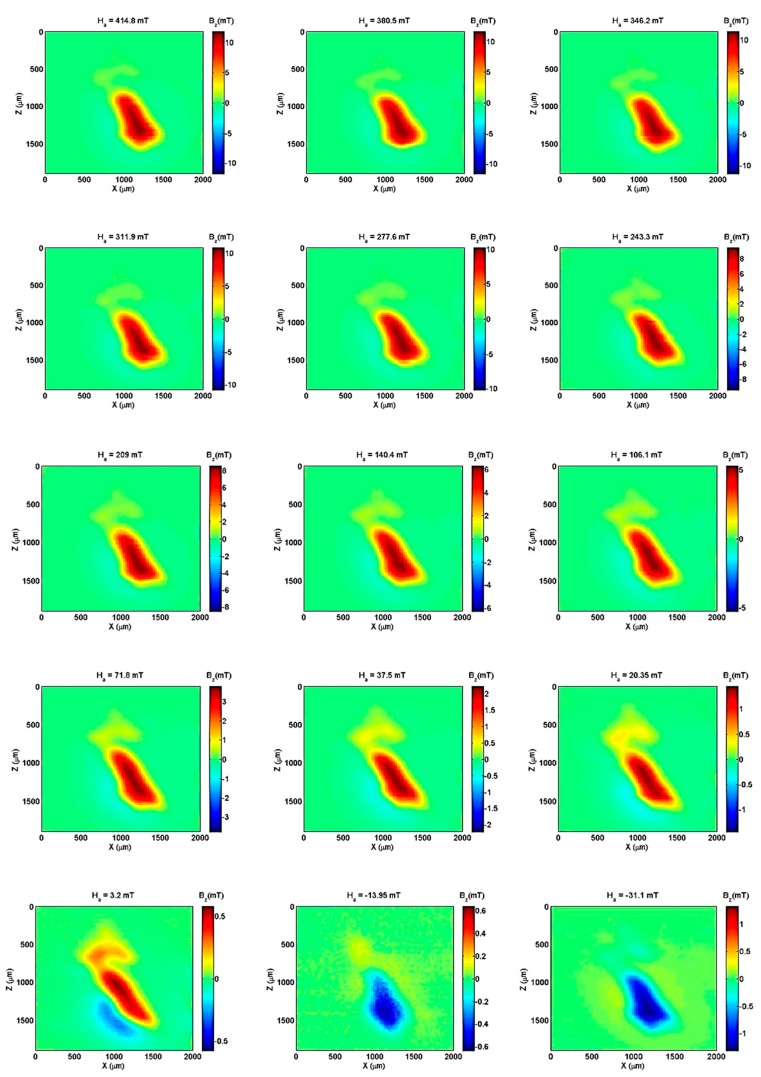
Experimental Bz maps (25 µm step) of the hook. Applied field varied from 415 mT to −31 mT.

**Figure 8 sensors-19-01636-f008:**
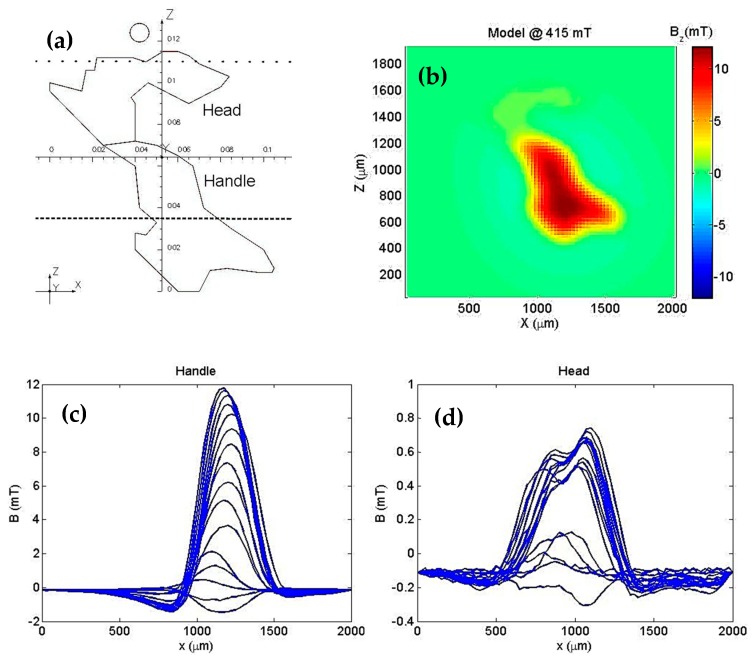
(**a**) Current loop model divided in two parts: the “handle” (high current) and “head” (low current). The scale is in cm. (**b**) Simulated B_y_ map with the current loop model. (**c**) B_y_ measured over the dashed line Z = 350 µm for different applied fields. (**d**) B_y_ measured over the dotted line Z = 1100 µm for different applied fields.

**Figure 9 sensors-19-01636-f009:**
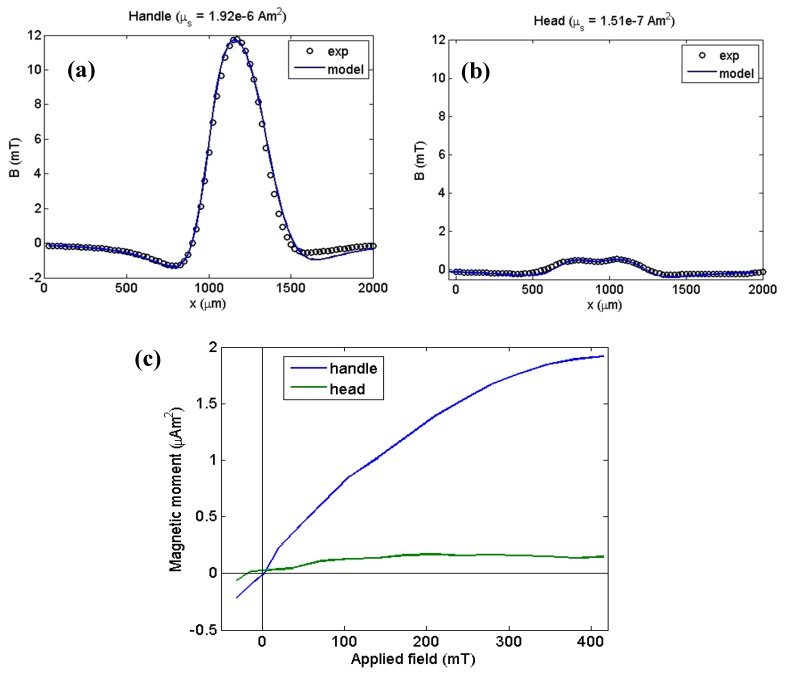
(**a**) Model fitting with experimental data over Z = 350 µm for the highest applied field. (**b**) Fitting with experimental data over Z = 1100 µm for the highest applied field. (**c**) Magnetization curves for the two parts of the hook.

**Figure 10 sensors-19-01636-f010:**
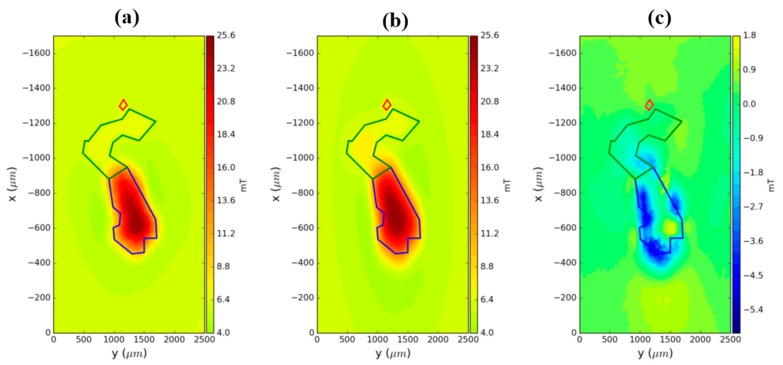
Comparison between synthetic data and observed data for the hook in Vredefort thin section. (**a**) Observed data from magnetic microscopy. (**b**) Synthetic data generated by model. (**c**) Residuals between panel a and b. The polygonal cross-section in blue is the handle part and the polygonal cross-section in green is the head part.

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
