# Peer review of "Characterizing Complex Mineral Structures in Thin Sections of Geological Samples with a Scanning Hall Effect Microscope"

_sensors, 2019, doi:10.3390/s19071636_

Reviewer 1 Report

The information provided in this manuscript will be very useful for the scientists working in this area. I have no hesitation to recommend publication of this work.

Only few questions:

1. The authors should introduce the possible mineral structure, is it only applicable to Jack Hills BIF sample?

2. What is the structure of this sample?

Author Response

Thank you for your evaluation of our article!

Reviewers' comments:

Reviewer #1:

1. The authors should introduce the possible mineral structure, is it only applicable to Jack Hills BIF sample?

We have presented two examples of different rock structures that reflect polyphasic histories of metamorphism and deformation. Our techniques can be applied to any type of rock (with magnetic carries) independent the complexity of its structures.

2. What is the structure of this sample?

This sample shows a micro-folded structure, represented by deformed micro-layers of iron oxide and silica hydroxides.

Author Response

Thank you for your evaluation of our article

Reviewers' comments:

Reviewer #2:

Dear Author,

 I have read the manuscript “Characterizing complex mineral structures in thin section of geological samples with a scanning Hall effect microscope.” The technique demonstrated is interesting and I’m especially impressed by the successful use of the gradiometer to reduce noise. I recommend publication after minor revisions.

My most important comment is that the manuscript does not discuss applications to geological problems adequately. An obvious area of application would be the measurement of hysteresis loops or at least coercivity spectra over a spatially-resolved sample. There are magnetic imaging devices with similar spatial resolution by much higher field and moment sensitivity (e.g., SQUID microscope). There are also devices with similar sensitivity but much higher spatial resolution (e.g., QDM). So this Hall effect sensor does not distinguish itself in these key areas. However, neither of the other instruments described can perform measurements under strong bias fields of ~0.5 T. The QDM is limited to about 0.04 T and the SQUID microscope is limited to even lower fields. If the authors can use this instrument at various applied fields to make even a minor loop in a hysteresis curve, that make the technique a lot more interesting for other users. The authors should discuss this potential.

We agree with the reviewer that this new type of magnetic microscopy can be used for an accurate investigation of magnetic mineralogy. However, we think that the specific use of the microscope for magnetic mineralogy research should make a separate study.

To further clarify the reviewer's remark, we have improved the text on lines 50-52, lines 62-63, lines 321-323.